# Effect of Bio-Fertilizer Application on Agronomic Traits, Yield, and Nutrient Uptake of Barley (*Hordeum vulgare*) in Saline Soil

**DOI:** 10.3390/plants13070951

**Published:** 2024-03-25

**Authors:** Mashael M. Alotaibi, Alya Aljuaid, Ibtisam Mohammed Alsudays, Abeer S. Aloufi, Aisha Nawaf AlBalawi, Abdulrahman Alasmari, Suliman Mohammed Suliman Alghanem, Bedur Faleh Albalawi, Khairiah Mubarak Alwutayd, Hany S. Gharib, Mamdouh M. A. Awad-Allah

**Affiliations:** 1Biology Department, College of Science and Humanities, Shaqra University, Shaqra 11961, Saudi Arabia; 2Department of Biology, College of Science, Qassim University, Burydah 52571, Saudi Arabia; 3Department of Biology, College of Science, Princess Nourah bint Abdulrahman University, P.O. Box 84428, Riyadh 11671, Saudi Arabia; 4Biology Department, University College of Haqel, University of Tabuk, Tabuk 71491, Saudi Arabia; 5Department of Biology, Faculty of Science, University of Tabuk, Tabuk 47512, Saudi Arabia; 6Department of Agronomy, Faculty of Agriculture, University of Kafrelsheikh, Kafrelsheikh 33516, Egypt; 7Field Crops Research Institute, Agricultural Research Center, Giza 12619, Egypt

**Keywords:** bio-fertilizers, salinity, NPK fertilizers, microbial activity, soil antioxidant, barley (*Hordeum vulgare*)

## Abstract

Under salinity conditions, growth and productivity of grain crops decrease, leading to inhibition and limited absorption of water and elements necessary for plant growth, osmotic imbalance, ionic stress, and oxidative stress. Microorganisms in bio-fertilizers have several mechanisms to provide benefits to crop plants and reduce the harmful effect of salinity. They can be effective in dissolving phosphate, fixing nitrogen, promoting plant growth, and can have a combination of all these qualities. During two successful agricultural seasons, two field experiments were conducted to evaluate the effect of bio-fertilizer applications, including phosphate solubilizing bacteria (PSB), nitrogen fixation bacteria and a mix of phosphate-solubilizing bacteria and nitrogen fixation bacteria with three rates, 50, 75 and 100% NPK, of the recommended dose of minimal fertilizer on agronomic traits, yield and nutrient uptake of barley (*Hordeum vulgare*) under saline condition in Village 13, Farafra Oasis, New Valley Governorate, Egypt. The results showed that the application of Microbein + 75% NPK recorded the highest values of plant height, spike length, number of spikes/m^2^, grain yield (Mg ha^−1^), straw yield (Mg ha^−1^), biological yield (Mg ha^−1^), protein content %, nitrogen (N), phosphorus (P), potassium (K) uptakes in grain and straw (kg ha^−1^), available nitrogen (mg/kg soil), available phosphorus (mg/kg soil), total microbial count of soil, antioxidant activity of soil (AOA), dehydrogenase, nitrogen fixers, and PSB counts. The application of bio-fertilizers led to an increase in plant tolerance to salt stress, plant growth, grain yield, and straw yield, in addition to the application of the bio-fertilizers, which resulted in a 25% saving in the cost of mineral fertilizers used in barley production.

## 1. Introduction

Barley (*Hordeum vulgare* L.) is one of the most important cereal crops. It is a dual-purpose crop, as it can be used as fodder and grain for animals, and is also one of the most important cover crops to enhance soil fertility. From a nutritional standpoint, its importance is because it is high in essential nutrients, protein, carbohydrates, fats and fiber [1]. Barley is a moderately salt-tolerant crop and has the ability to be cultivated under a broad range of ecological stresses, including saline, drought, and heating conditions [2,3]. In arid and semi-arid regions, salinity is one of the main abiotic stresses that negatively influence crop productivity and quality [4]. In Egypt, the total salt-affected area is about 0.9 M ha [5].

Salinity stress leads to many effects on the physiological state of the plant. It may prevent or hinder various physiological processes, from absorption of nutrients to inhibition of DNA replication and the biosynthesis of large molecules. This leads to changes in metabolism, growth, and development, and thus it leads to decreased productivity [6,7,8].

In recent years, the use of chemical fertilizers in agriculture has led to an increase in the rate of self-sufficiency in food production in the world, but this has been at the expense of the ecosystem and the safety of all living organisms [9,10]. Excessive use of chemical fertilizers over the past few decades has led to numerous negative impacts on soil quality and fertility and increased suitability of agricultural resources, such as increased soil salinity and negative impacts on the ecosystem and human health [11,12]. Therefore, reducing the use of chemical fertilizers in agricultural areas is of huge importance in order to protect plant health, reduce environmental pollution, and reduce production costs. To meet agricultural production requirements, beneficial microorganisms are the best alternatives to traditional farming methods. Microorganisms in the soil are very essential for the growth and development of plants. They play a major role in soil fertility, increasing the content and biodiversity of the soil and facilitating nutrients for the plant [9,10]. Therefore, the importance of using bio-fertilizers with microorganisms such as phosphate-dissolving bacteria, fixing nitrogen, and plant growth-promoting rhizobacteria (PGPR) appears in order to increase productivity and preserve the environment. The use of these microorganisms represents environmentally friendly alternatives, and plant growth-promoting rhizobacteria (PGPR) are free-living bacteria in the roots of plants that colonize them [13,14]. Moreover, they are incorporated into many bioactivities of the soil system to make them active for nutrient movement and viable for crop production [15]. In addition, through its ability to fix N_2_, it works to enhance and promote plant growth, soluble mineral phosphate, and other nutrients, regulate soil nutrients, synthesize various plant growth hormones, protect plants from plant pathogens, improve soil structure, and mitigate the effect of abiotic stress, e.g., salinity, drought, acidity, humidity, bioremediation of different classes of toxic heavy metals and degradation of synthetic chemical compounds [16,17,18,19,20]. In addition to the previously mentioned functions of PGPRs, they also play a major role in plant defense in various ways by producing antibiotics, volatiles, bio-surfactants, siderophores, and enzymes that denature the cell wall and bring systemic resistance [21,22]. Biologically, Azotobacter and Azospirillum are aerobic heterotrophs that have the ability to fix atmospheric nitrogen and make it available to plants. Azotobacter also secretes some hormones such as gibberellin, thiamine, indole acetic acid, and some sugars that work to improve soil properties [23,24,25].

Bio-fertilizers are distinguished from chemical fertilizers in that they are considered safer than chemical fertilizers for various reasons, including that they are less expensive, they cause less environmental damage, their activity is more concentrated, and they are more efficient when used in smaller quantities [10,22,26]. Moreover, they also have the ability to reproduce while being simultaneously regulated by the plant and local microbes. Furthermore, microbial inoculants have faster decomposition processes and do not cause an increase in pathogens and pests to develop tolerance [10,22,27,28].

An additional advantage of bio-inoculators is that they are highly efficient and can be used many times without any harmful effect on plant productivity. Bio-fertilizers play a major role with organic and chemical fertilizers, in increasing organic carbon in the soil, supporting sustainability in agricultural production, and meeting the plant’s need for mineral nutrients [22,28,29,30,31]. The expression bio-fertilizers refers to a broad range of products that contain live or inactive microorganisms, including bacteria, algae, fungi, and actinomycetes. These products are used and applied for many purposes. They work to fix atmospheric nitrogen, dissolve and mobilize soil nutrients, or one of these processes, in addition to secreting substances that promote plant growth [32,33,34]. Currently, bio-fertilizers are available as alternatives to traditional chemical fertilizers or are used to reduce the amounts of chemical fertilizers used [35].

The production of bio-fertilizers is increasing year after year, and the size of the bio-fertilizer market is expected to increase and expand to include larger areas. The world’s bio-fertilizer market value reached USD 3.14 billion in 2022, and its value is expected to reach USD 5.2 billion by 2032, at a compound annual growth rate (CAGR) is 11.3% between 2023 and 2032 [31,36]. Therefore, many researchers have attempted to reduce the doses used for chemical fertilizers by using bio-fertilizers in different treatments containing both gradual and varied proportions [32].

Nitrogen is one of the important nutrients for plants, as they need it in different stages of growth. It is also an important part of amino acids, proteins, enzymes, and protoplasm. In addition, N is part of a cofactor such as NAD and NADH in amino acid formation reactions in cells [10,37,38,39,40]. Phosphorus is an essential and main element in plant nutrition, as it plays an important role in the processes of growth, formation, and division of cells, the formation and development of seeds, and in the synthesis of energy compounds such as CTP, ATP, and GTP [39,40,41,42]. As for potassium, it plays a major role in the processes of plant growth and development, as it is of great importance in the process of photosynthesis. It also works to increase cell division and the plant’s resistance to abiotic and biotic stresses [43,44,45,46].

Thus, the present study was conducted with the objective of studying the impact of bio-fertilizer treatments and three diverse levels of mineral fertilizer (50, 75, and 100%) of the recommended dose on the growth, grain yield, productivity, and chemical composition of barley (*Hordeum vulgare* L) under salinity conditions in newly reclaimed lands and attainment a treatment to reduce the amounts of mineral fertilization used and, thus, reduce the expenditure of production and improve the environment.

## 2. Results

### 2.1. The Effect of Bio-Fertilizers and Mineral Fertilizer Rates on the Agronomic Traits of Barley Plants under Saline Conditions

The treatment by bio-fertilizers with rates of the recommended dose of mineral fertilizers under salinity stress conditions gave the highest values of plant height (PH), spike length (Sp L), spike weight (Sp W), grains spike (G/Sp), 1000-grain weight (1000-GW), grain yield (GY) (Mg ha^−1^), Straw Y (Mg ha^−1^), and biological Y (Mg ha^−1^), at the second and first year, respectively, (Table 1 and Figure 1).

The combinations of Microbein + 75% NPK (T8), followed by Rhizobacterin + 75% NPK (T7), and then next by fertilization with Microbein + 100% NPK (T12), gave the highest values for the agronomic traits (Table 1 and Figure 1).

The data showed that significant differences were obtained in plant height, spike length, and spike weight of barley between the treatments in both studied seasons, as shown in Table 1. In this trend, the tallest plants were obtained with treatments T8, T12, and T7, followed by Microbein + 50% NPK (T4), and the increasing percentage was by 32.7, 30.7, 23.2, and 14.8% in the first year, while the increase in second was by 28.8, 24.8, 20.9, and 22.4% over the recommended dose of mineral fertilizer. On the other hand, the increase over the same dose of mineral fertilizer application treatments were 34.3, 30.7, 24.7, and 42.3 and 33.8, 24.8, 25.6, and 56.4% in the first and second years, respectively.

The highest values of spike length were recorded at harvest with the applied parameters of T8, T12, and T7, with an increase of 32.3, 30.3, and 23.2 and 28.9, 25.0, and 21.2% compared with the recommended dose of mineral fertilizers 100% NPK (T9) in both seasons, respectively, while the increase over the same dose of mineral fertilizer application treatments without bio-fertilizer were 33.7, 30.3, 24.5, and 34.0, 25.0, and 26.0% in the first and second year, respectively (Table 1).

Rhizobacterin + 100% NPK (T11), T12, and T8 T7 achieved the heaviest weight of spike. The increase in the spike weight compared with a recommended dose of mineral fertilizer without bio-fertilizer (T9) was by the percentage of 32.3, 32.3, 29.0, and 22.6 and 35.5, 35.5, 32.3, and 29.0%, in the first and second year, while the percentage increase compared with the same dose of mineral fertilizer without bio-fertilizer was by 32.3, 32.3, 73.9, and 65.2 and 35.5, 35.5, 78.3, and 73.9% in the first and second year, respectively (Table 1).

For the number of grains spike^−1^, T12, T11, T8, and T7 gave the highest values at the first year and second year, respectively, as shown in Table 1, with an increased percentage over the recommended dose of mineral fertilizer without bio-fertilizer (T9): 35, 34.6, and 77 and 72.1, 36.1, 35.9, 29.1, and 25.5%, at the first year and second year, respectively. Meanwhile, the increase percentages over the same dose of mineral fertilizer without bio-fertilizer were 35, 34.6, 74.3, and 69.5, and 36.1, 35.9, 77, and 72.1%, in the first year and second year, respectively, as shown in Table 1.

With regard to the number of spikes m^−2^, the treatments T8, T12, T7, T11, and Phosphorein + 75% NPK (T6) gave the highest values at the first year and second year, respectively, as shown in Table 1, with increased percentages over the recommended dose of mineral fertilizer without bio-fertilizer (T9): 23.4, 18.9, 16.8, 15.6, and 13.1 and 50.0, 39.5, 36.6, 31.1, and 28.5% at the first year and second year, respectively. Meanwhile, the increased percentages compared to the same dose of mineral fertilizer without bio-fertilizer were 19.5, 18.9, 15.6, 13.1, and 11.1 and 59.0, 39.5, 31.1, 44.9, and 23.6% in the first year and second year, respectively, as shown in Table 1.

Concerning 1000-grain weight (g), T12, T11, T8, T7, and Phosphorein + 100% NPK (T10) gave the highest values at the first year and second year, respectively, as shown in Table 1. The percentage increases over the recommended dose (T9) were 34.8, 33.8, 29.0, 25.5, and 21.3 and 36.3, 36.1, 31.8, 28.1, and 24.3% in the first year and second year, respectively. Meanwhile, the increases over the same dose of mineral fertilizer without bio-fertilizer were 34.8, 33.8, 74.9, and 70.1 and 36.3, 36.1, 77.2, and 72.4% in the first year and second year, respectively, as shown in Table 1.

Regarding grain yield (Mg ha^−1^), T8, T7, and T12 gave the highest values of Mg ha^−1^ at the first year and second year, respectively, followed by T4, T11, T10, T6, Rhizobacterin + 50% NPK (T3), and Phosphorein + 50% NPK (T2), which gave the values at the first year and second year, respectively, as shown in Figure 1. The percentage increase compared with the recommended dose (T9) were 51.3, 46.2, 43.6, 38.5, 38.5, 35.9, 35.9, 33.3, and 30.8 and 62.5, 52.5, 55.0, 37.5, 42.5, 37.5, 35.0, 35.0, and 30.0% in the first year and second year, respectively. Meanwhile, the increases compared with the same dose of mineral fertilizer without bio-fertilizer were 47.5, 40.0, 46.2, 58.8, 38.5, 35.9, 32.5, 52.9, and 50.0 and 58.5, 51.2, 52.5, 48.77, 42.5, 35.0, 34.2, 46.0, and 40.5% in the first year and second year, respectively, as shown in Figure 1.

For straw yield (Mg ha^−1^), T8, T12, T7, T4, T11, T6, T10, T3, T2, and without bio-fertilizer + 75% NPK (T5) gave the highest values at the first year and second year, respectively, as shown in Figure 1. The percentage increases compared with the recommended dose (T9) were 53.9, 48.1, 44.2, 40.4, 40.4, 38.5, 36.5, 32.7, and 30.8 and 59.3, 50.0, 53.7, 37.0, 40.7, 37.0, 33.3, 33.3, and 27.8% in the first year and second year, respectively. Meanwhile, the increases over the same dose of mineral fertilizer without bio-fertilizer were 50.9, 48.1, 41.5, 62.2, 40.4, 35.9, 36.5, 53.3, and 51.1 and 56.4, 50.0, 50.9, 51.0, 40.7, 34.6, 33.3, 46.9, and 40.8% in the first year and second year, respectively, as shown in Figure 1.

Concerning biological yield (Mg ha^−1^), T8, T12, T7, T4, T11, T6, T10, T3, T2, and T5 gave the highest values at the first year and second year, respectively, as shown in Figure 1. The percentage increases compared with the recommended dose (T9) were 52.8, 48.4, 42.9, 39.6, 39.6, 37.4, 35.2, 33.0, 30.8, and 2.2 and 60.6, 51.1, 53.2, 37.2, 41.5, 37.2, 34.0, 34.0, 28.7, and 3.2% in the first year and second year, respectively. In the meantime, the increases over the same dose of mineral fertilizer without bio-fertilizer were 49.5, 48.4, and 39.8 and 60.8, 39.6, 34.4, 35.2, 53.2, 50.6, 55.7, 51.1, 48.5, 50.0, 41.5, 33.0, 34.0, 46.6, and 40.7%, in the first year and second year, respectively, as shown in Figure 1.

### 2.2. Effect of Bio-Fertilizers and Mineral Fertilizer Levels on NPK Uptake and Protein Contents

In relation to protein content %, T12, T11, T8, T7, and T10 gave the highest values at the first year and second year, respectively, as shown in Figure 2. The percentage increases compared with the recommended dose (T9) were 39.6, 37.8, 28.8, 25.2, and 21.6 and 41.4, 39.6, 32.4, 27.9, and 24.3% in the first year and second year, respectively. In the interim, the increases compared with the same dose of mineral fertilizer without bio-fertilizer were 39.6, 37.8, 74.4, 69.5, and 21.6 and 41.4, 39.6, 77.1, 71.1, and 24.3% in the first year and second year, respectively, as shown in Figure 2.

Concerning grain N uptake (kg ha^−1^), T12, T11, T8, T7, and T10 gave the highest values at the first year and second year, respectively, as shown in Figure 2. The percentage increases compared with the recommended dose (T9) were 41.0, 38.5, 30.0, 26.4, and 22.3 and 41.1, 39.6, 31.8, 28.0, and 24.3% in the first year and second year, respectively, while, the increases over the same dose of mineral fertilizer without bio-fertilizer were 41.0, 38.5, 74.6, 69.7, and 22.3 and 41.1, 39.6, 77.2, 72.2, and 24.3% in the first year and second year, respectively, as shown in Figure 2.

For grain K uptake (kg ha^−1^), T8, T12, T6, T7, T10, T11, T4, and T3 gave the highest values at the first year and second year, respectively, as shown in Figure 2. The percentage increases compared with the recommended dose (T9) were 51.3, 45.5, 40.0, 37.1, 37.1, 34.2, 20.0, and 2.9 and 89.5, 86.7, 53.9, 50.0, 47.2, 44.8, 21.0, and 10.5% in the first year and second year, respectively. Meanwhile, the increases compared with the same dose of mineral fertilizer without bio-fertilizer were 60.6, 45.5, 48.7, 45.6, 37.1, 34.2, 55.7, and 33.5 and 95.0, 86.7, 58.3, 54.3, 47.2, 44.8, 53.1, and 39.8% in the first year and second year, respectively, as shown in Figure 2.

Concerning grain P uptake (kg ha^−1^), T8, T12, T6, T10, T7, T11, T4, and T3 gave the highest values at the first year and second year, respectively, as shown in Figure 2. The percentage increases compared with the recommended dose (T9) were 51.9, 50.0, 46.2, 41.5, 37.7, 34.0, 19.8, and 2.8 and 50.0, 47.3, 45.5, 38.2, 37.3, 33.6, 19.1, and 10.9% in the first year and second year, respectively, while the increases over the same dose of mineral fertilizer without bio-fertilizer were 61.0, 50.0, 55.0, 41.5, 46.0, 34.0, 54.9, and 32.9 and 58.7, 47.3, 53.9, 38.2, 45.2, 33.6, 50.6, and 40.2% in the first year and second year, respectively, as shown in Figure 2.

Regarding straw N uptake (kg ha^−1^), T12, T11, T8, T7, and T10 gave the highest values at the first and second year, respectively, as shown in Table 2. The percentage increases compared with the recommended dose (T9) were 39.1, 36.8, 28.5, 25.0, and 20.9 and 40.1, 38.7, 31.1, 27.4, and 23.7% in the first year and second year, respectively. At the same time, the percentage increases compared with the same dose of mineral fertilizer without bio-fertilizer were 39.1, 36.8, 72.6, 67.9, and 20.9 and 40.1, 38.77, 75.1, 70.2, and 23.7% in the first year and second year, respectively, as shown in Table 2.

Regarding straw K uptake (kg ha^−1^), T4, T12, T8, T10, T11, and T7 gave the highest values at the first year and second year, respectively, as shown in Table 2. The percentage increase compared with the recommended dose (T9) were 19.9, 39.1, 28.5, 20.9, 36.8, and 25.0 and 19.9, 40.1, 31.1, 23.7, 38.7, and 27.4% in the first year and second year, respectively. In the interim, the increases over the same dose of mineral fertilizer without bio-fertilizer were 58.7, 17.7, 15.3, 6.4, 3.4, and 7.6 and 58.9, 17.7, 15.0, 6.5, 3.3, and 7.3% in the first year and second year, respectively, as shown in Table 2.

For straw P uptake (kg ha^−1^), T8, T12, T6, T10, T7, T11, T4, T2, and T3 gave the highest values at the first year and second year, respectively, as shown in Table 2. The percentage increases compared with the recommended dose (T9) were 52.7, 50.0, 49.1, 42.9, 39.39, 36.6, 19.6, 8.0, and 0.0 and 50.0, 44.8, 44.8, 36.2, 36.2, 30.2, 18.1, 8.6, and 4.3% in the first year and second year, respectively. In the meantime, the increases compared with the same dose of mineral fertilizer without bio-fertilizer were 64.4, 50.0, 60.6, 42.9, 50.0, 36.6, 55.8, 40.7, and 30.2 and 58.2, 44.8, 52.7, 36.2, 43.6, 30.2, 53.9, 41.6, and 36.0% in the first year and second year, respectively, as shown in Table 2.

### 2.3. Presence of Minerals Available in the Rhizosphere of Barley Plants under Saline Conditions

Regarding available nitrogen (mg/kg soil), the application of bio-fertilizer combined with rates of mineral fertilizer significantly improved the availability of nitrogen in the soil, and the treatments T8, T12, T6, T10, T7, T11, T4, T2, and T3 gave the highest values at the first year and second year, respectively, as shown in Figure 3. The percentage increases compared with the recommended dose (T9) were 71.7, 67.9, 51.6, 51.6, 34.0, 28.3, 15.7, 8.8, and 2.5 and 72.5, 61.1, 53.3, 45.5, 29.3, 25.2, 16.2, 4.8, and 0.0% in the first year and second year, respectively. Meanwhile, the increases compared with the same dose of mineral fertilizer without bio-fertilizer were 76.1, 67.9, 55.5, 51.6, 37.4, 28.3, 84.0, 73.0, and 63.0 and 78.9, 61.1, 59.0, 45.5, 34.2, 25.2, 81.3, 63.6, and 56.1% in the first year and second year, respectively, as shown in Figure 3.

For available phosphorus (mg/kg soil), the application of treatments of bio-fertilizer combined with rates of mineral fertilizer significantly improved the availability of phosphorus in the soil, and the treatments T8, T12, T7, T6, T11, T10, T4, and T2 gave the highest values at the first year and second year, respectively, as shown in Figure 3. The percentage increases compared with the recommended dose (T9) were 53.7, 50.8, 44.8, 43.3, 35.8, 32.8, 17.9, and 4.5 and 52.8, 48.6, 29.2, 41.7, 30.6, 30.6, 16.7, and 2.8% in the first year and second year, respectively. In the same direction, the increase over the same dose of mineral fertilizer without bio-fertilizer were 58.5, 50.6, 49.2, 47.7, 35.8, 32.8, 51.9, and 34.6 and 59.4, 48.6, 34.8, 47.8, 30.6, 30.6, 52.7, and 34.6% in the first year and second year, respectively, as shown in Figure 3.

### 2.4. Biological Activities in the Rhizosphere of Barley Plants under Salinity-Affected Soil Conditions

Concerning total microbial count (×10^5^ cfu/g dry soil) was greatly affected by the application of bio-fertilizer, the treatments T8, T7, T12, T11, T6, T10, T4, T3, and T2 gave the highest values at the first year and second year, respectively, as shown in Figure 3. The percentage increase compared with the recommended dose (T9) were 361.8, 345.5, 302.5, 262.9, 185.8, 169.1, 107.3, 78.7, and 40.5 and 394.8, 370.4, 331.3, 283.3, 202.2, 182.6, 124.1, 92.6, and 51.1% in the first year and second year, respectively. Meanwhile, the increases compared with the same dose of mineral fertilizer without bio-fertilizer were 350.5, 334.6, 302.5, 262.9, 178.8, 169.1, 135.3, 102.9, and 59.5 and 341.1, 319.4, 331.3, 283.3, 169.4, 182.6, 135.9, 102.8, and 59.1% in the first year and second year, respectively, as shown in Figure 3.

Regarding antioxidant activity of soil ug ascorbic acid/g dry soil (AOA), T8, T7, T12, T11, T6, T10, T4, T3, and T2 have the maximum impact on increasing the antioxidant activity of soil in both seasons with values in the first year and second year, respectively, as shown in Figure 3. The percentage increases compared with the recommended dose (T9) were 144.4, 133.3, 122.2, 100.0, 50.0, 38.9, 27.8, 22.2, and 5.6 and 142.1, 131.6, 126.3, 115.8, 47.4, 36.8, 26.3, 21.1, and 15.8% in the first year and second year, respectively. Meanwhile, the increases compared with the same dose of mineral fertilizer without bio-fertilizer were 193.3, 180.0, 122.2, 100.0, 80.0, 38.9, 91.7, 83.3, and 58.3 and 187.5, 175.0, 126.3, 115.8, 75.0, 36.8, 84.6, 76.9, and 69.2% in the first year and second year, respectively, as shown in Figure 2.

Concerning dehydrogenase (enzyme μg TPF/g dry soil/24 h), results indicated that the treatments T7, T8, T11, T12, T3, T6, T4, T10, and T2 obtained the highest values at the first year and second year, respectively, as shown in Figure 3. The percentage increases compared with the recommended dose (T9) were 383.6, 347.5, 337.7, 290.2, 268.9, 239.3, 234.4, 221.3, and 167.2 and 381.0, 344.4, 334.9, 287.3, 266.7, 236.5, 231.6, 219.1, and 165.1% in the first year and second year, respectively, while the increases compared with the same dose of mineral fertilizer without bio-fertilizer were 383.6, 347.5, 337.7, 290.2, 341.2, 239.3, 300.0, 221.3, and 219.6 and 381.0, 344.4, 334.9, 287.3, 344.2, 236.5, 301.9, 219.1, and 221.2% in the first year and second year, respectively, as shown in Figure 3.

Regarding nitrogen fixers (MPN/gm dry soil) 10^3^, the application of treatments of T8, T7, T12, T10, T11, T6, T4, T3, and T2 showed the highest values at the first year and second year, respectively, as shown in Figure 3. The percentage increases compared with the recommended dose (T9) were 412.1, 409.2, 312.1, 273.8, 264.8, 256.2, 111.8, 103.2, and 76.7 and 382.9, 382.9, 294.0, 255.2, 249.7, 241.3, 105.4, 94.3, and 72.0% in the first year and second year, respectively. At the same time, the increases over the same dose of mineral fertilizer without bio-fertilizer were 427.3, 424.3, 312.1, 273.8, 264.8, 266.8, 148.3, 138.2, and 107.1 and 382.9, 382.9, 294.0, 255.2, 249.7, 241.3, 138.5, 125.6, and 99.7% in the first year and second year, respectively, as shown in Figure 3.

The phosphate-solubilizing bacteria (PSB) count (CFU/gm of dry soil) was significantly affected by the use of bio-fertilizers in the application of treatments T10, T6, T8, T7, T12, T11, T2, T4, and T3 to obtain the highest values at the first year and second year, respectively, as shown in Figure 3. The percentage increases compared with the recommended dose (T9) were 497.6, 459.8, 447.6, 373.2, 348.8, 248.8, 174.4, 124.4, and 86.6 and 443.5, 421.7, 410.9, 332.6, 300.0, 221.7, 155.4, 121.7, and 89.1% in the first year and second year, respectively. Meanwhile, the increases compared with the same dose of mineral fertilizer without bio-fertilizer were 497.6, 546.5, 532.4, 446.5, 348.8, 248.8, 268.9, 201.6, and 150.8 and 443.5, 485.4, 473.2, 385.4, 300.0, 221.7, 231.0, 187.3, and 145.1% in the first year and second year, respectively, as shown in Figure 3.

## 3. Discussion

### 3.1. The Effect of Bio-Fertilizers and Mineral Fertilizer Rates on the Agronomic Traits of Barley Plants under Saline Conditions

Barley is considered a type of crop that tolerates salt, but barley productivity is affected by salinity stress and osmotic pressures. Salt stress leads to a decrease in the germination rate and all growth parameters in barley. Treatment and inoculation with PGP bacteria led to enhancing and increasing plant growth under salinity conditions through the production of growth-promoting nutrients and growth regulators [47].

The results in Table 1 and Table 2 and Figure 1, Figure 2 and Figure 3 showed that the application of a combination of bio-fertilizers with levels of mineral fertilizers led to a significant increase in productivity and yield contributing traits of barley plants growing under saline stress.

The results of the current study showed that the most of studied traits showed the highest values when applying the treatments T8, T7, T12, especially grain yield (Mg ha^−1^) and some related traits, or it may be assigned T12, higher than T8, with a little and insignificant difference. This indicates the importance of applying bio-fertilizer in reducing NPK mineral fertilizer doses by 25% without affecting the yield and other characteristics or their decrease (Table 1 and Table 2, and Figure 1, Figure 2 and Figure 3). The inoculation by bio-fertilizer significantly improved agronomic traits and yield components than the recommended dose of mineral fertilizer without bio-fertilizer (T9) treatment, and compared with the same dose of mineral fertilizer without bio-fertilizer. Under saline conditions, bio-fertilizers promote plant growth by facilitating enhanced nutrients, producing growth-promoting substances and growth regulators [22,48,49].

The reason for the increase in yield and its components in treatments received with bio-fertilizers is mainly due to the beneficial effect of applying bio-fertilizers to the soil, as this improved the biological and chemical properties of the soil. All of these effects resulted in an increase in the availability and release of more nutrients available to the root plants [50,51,52].

The results showed that bio-fertilizers have the ability to enhance plant growth, due to their ability to solubilize phosphate and produce organic acids, exopolysaccharides, and IAA. Therefore, microorganisms with PGP activities can lessen biotic and abiotic stresses and are thus useful in agricultural fields [53,54,55]. PGP microbes improve stress tolerance in plants through various mechanisms, such as the production of antioxidants that can detoxify reactive oxygen species, which leads to improved growth standards, the production of growth regulators, and, most importantly, increased nutrient content through nitrogen fixation and dissolution of elements. All of these mechanisms lead to improving plant health under salt stress [47,56,57].

### 3.2. Effect of Bio-Fertilizers and Mineral Fertilizer Levels on NPK Uptake and Protein Contents

The results in (Figure 2 and Table 2) showed that the effects of applying the three bio-fertilizers containing N_2_-fixing and P-solubilizing PGPR strains led to a significant increase in the grain and straw contents of nitrogen, phosphorus, potassium, as well as protein in the barley plants under study. The highest values of N, P, K, and protein % contents were obtained from Microbein bio-fertilizer inoculation (mixed strains) combined with 75% NPK level, which increased N contents of grain by 40.64 and 40.64% P content in the two seasons, respectively. At the same time, the highest values of N, P, and K uptake were obtained from T8, which increased N contents of straw in the two seasons, respectively. A significant increase was recorded in the levels of nitrogen, phosphorus, and potassium when applying bio-fertilized to barley, corn, and wheat plants compared with the recommended dose (control). A significant increase was also recorded and observed in the percentage of water- and salt-soluble protein fractions (albumin and globulin) when applying bio-fertilization [58,59,60].

### 3.3. Presence of Minerals Available in the Rhizosphere of Barley Plants under Saline Conditions

One of the negative effects of high salinity in natural conditions is that it greatly affects microbial communities in the soil and prevents the mineralization of organic matter in the soil. This leads to sharp changes in the process of turnover of organic substances [61], thus reducing the release of plant nutrients. Here comes the importance of applying bio-fertilizers to the soil under conditions of salinity stress to decrease the damaging effects of salinity and increase the number of beneficial microbes in the soil. The data in Figure 3, revealed that the application of T8 contained bio-fertilizer combined with 75% mineral fertilizer significantly improved the availability of nitrogen and phosphorus in soil under the current study (Figure 3). Application of all treatments containing bio-fertilizers gave values of available nitrogen and phosphorus much higher than the control treatment (T9), which is the recommended rate of mineral fertilizers without adding bio-fertilizers. Application of T8 gave the highest values of available nitrogen and phosphorus over the control at the studied season, respectively, as shown in Figure 3. On the other hand, the application of bio-fertilizer gave values of available nitrogen and phosphorus much higher than the same dose of mineral fertilizer without bio-fertilizer, over the same dose of mineral fertilizer without bio-fertilizer at the studied season, respectively, as shown in Figure 3.

Bio-fertilizers containing PGPR bacteria play a necessary role in providing and cycling nutrients in the soil through several processes, including that they participate in processes such as nitrogen fixation, nitrification, oxidation, ammonification, and other processes that lead to the decomposition of organic matter in the soil and the release of vital inorganic plant nutrients in the soil [62,63,64]. Kurokura et al. [65] show that the secretion of organic acids by PGPR contributes significantly to increasing the solubility of phosphorus and converting it from insoluble to soluble forms [66,67,68].

The availability of essential nutrients in the soil is very important for plants, as they mainly depend on them for their development and growth, which serves as the main reservoir through which plants obtain the nutrients necessary for their growth and development. However, the majority of nutrients in the soil are in insoluble and unabsorbable forms, making the availability of these nutrients restricted for plants. Whereas mineral fertilization is considered an effective and quick method, it is not currently recommended due to its harmful effects on the ecosystem and soil [69,70]. On the other side, over time, bio-fertilizers have proven to be an effective means of providing the necessary nutrients to plants and maintaining the sustainability of the agricultural system and the ecosystem in general. Bio-fertilizers consist of a variety of beneficial microorganisms that are able to decompose essential nutrients from insoluble compounds, making them available to plants. As a result of successive studies on this, there are a large number of microorganisms, including bacteria, fungi, and actinomycetes, which have properties that enhance the dissolution of metal ions through various mechanisms, such as changing the pH of the soil or direct heavy metal removal from metal cations [69,70,71].

One of the useful indicators for knowing the biological condition and quality of the soil is to rely on the antioxidant system to determine this [72]. Increased antioxidant activity in the soil protects the plant from oxidative stress resulting from high salinity [6]. Many researchers have also confirmed that there is a close correlation between the total antioxidant capacity of soil and the number of microbes in different soil types. In addition, enzymatic and non-enzymatic antioxidants play a vital function in the development of plants and growth, especially in protection mechanisms. Protracted exposure to ecological stress leads to oxidative stress, causing a pathological condition [73]. Several studies reported that peroxidases and phenolic compounds contributed to protecting the plant from oxidative stress [74,75]. High salinity affects the soil in many ways, including its effect on microbial communities and their growth in the soil. It also prevents the mineralization of organic matter in the soil, which results in severe changes in the rate of organic matter turnover, thus reducing the release of plant nutrients [49,76]. The results of the current study showed that the application of bio-fertilizer with mineral fertilizer levels led to a significant enhancement in the availability of nitrogen and phosphorus in the soils examined (Figure 3).

### 3.4. Biological Activities in the Rhizosphere of Barley Plants under Salinity-Affected Soil Conditions

The numbers of microorganisms, or what is known as microbial density, in the soil are affected by the presence of mineral fertilizers levels, mainly NPK, as well as the levels of applied bio-fertilizers, as shown in Figure 3. Microbial density increases with the addition of mineral fertilizers as well as bio-fertilizers, but it increases to a certain limit and is not a continuous increase. The high dose of mineral fertilizers used in agricultural systems may lead to restricting the growth of microbes and their activities in the soil [77,78]. In the same direction, with regard to the effect of bio-fertilizers, the results showed an increase in microbial density when adding bio-fertilizers in the presence of the same rate of mineral fertilizer, with a relative decrease in the greater rate of 100% NPK. The results indicated that the total number of microbes was higher when 75% NPK/ha was applied than that obtained at 50% NPK/ha or 100% NPK/ha. Several studies have revealed that microbial community structure, microbial activity, and soil microbial biomass are decreased by high salinity [79,80]. However, the use of halo-tolerant bacteria with PGP properties enhanced soil microbial biomass through osmolyte production [81,82,83].

The antioxidant system is a very useful indicator of soil quality and the biological condition of the soil [84,85,86]. The dehydrogenase enzyme increases with the addition of bio-fertilizers with the increase in the addition of mineral fertilizers, but it increases to a certain extent and not a continuous increase. In the same direction and with regard to the effect of bio-fertilizers, the results showed an increase in the dehydrogenase enzyme when adding bio-fertilizers in the presence of the same rate of mineral fertilizers, with a relative decrease in the rate greater than 100%. The results indicated that the dehydrogenase enzyme was higher when using 75% NPK/ha compared to that obtained at 50% NPK/ha or 100% NPK/ha (Figure 3). Increased antioxidant activity in the soil protects the plant from oxidative stress resulting from high salinity [87,88]. Enzymatic and non-enzymatic antioxidants play an essential function in plant development and growth, especially in protection mechanisms [89,90,91]. The growth of plants under environmental stresses leads to oxidative stress, causing a pathological condition. Numerous studies have confirmed the existence of a strong relationship between the number of microbes and the total antioxidant capacity in the soil and different soil types [92,93].

## 4. Materials and Methods

### 4.1. The Experiment Site

During the two successive seasons in the years 2019 and 2020 in new reclaimed sandy salt-affected soil at Village 13, Farafra Oasis (Latitude: 27°3′24″ N; Longitude: 27°58′11″ E), New Valley Governorate, Egypt, two field experiments were conducted to study the response of barley plant (*Hordeum vulgare*, L. c.v. Giza-123) to the three recommended levels of mineral fertilizer NPK (50, 75, and 100%) and inoculation by Azotobacter chrococcum (in the form of a commercial bio-fertilizer compound for nitrogen fixation, Rhizobacterin), Bacillus subtilis as phosphate-solubilizing bacteria (PDB) (in the form of a commercial bio-fertilizer compound to fix nitrogen, Phosphorein), mixture of Pseudomonas, Azotobacter, Bacillus, and Rhizobium (in the form of a commercial bio-fertilizer compound for nitrogen fixation, Microbein), and without inoculation as control. The three bio-fertilizers were used at the rate of Rhizobacterin 2.4 kg, Phosphorien 1.4 kg, and Mycrobein 1 kg per 143 kg barley grains ha^−1^. Inoculation of barley grains with Rhizobacterin, Phosphorien, and Mycrobein was carried out after coating the grains with an adhesive material (gum arabic 5%), then planting and irrigating them immediately after inoculation. The bio-fertilizers were obtained from the General Organization for Agricultural Equalization Fund (GOAEF), Egypt’s Ministry of Agriculture and Land Reclamation. The main properties regarding soil and water used in the experiment are detailed in Table 3.

### 4.2. Experimental Design and Management

Over 2 years, 2 field experiments with 12 treatments were conducted in 3 replications in a randomized complete block design (RCBD). The treatments included without bio-fertilizer + 50% NPK (T1), Phosphorin + 50% NPK (T2), Rhizobacterin + 50% NPK (T3), Microbein + 50% NPK (T4), without bio-fertilizer + 75% NPK (T5), Phosphorin + 75% NPK (T6), Rhizobacterin + 75% NPK (T7), Microbein + 75% NPK (T8), without bio-fertilizer + 100% NPK (T9), Phosphorin + 100% NPK (T10), Rhizobacterin + 100% NPK (T11), and Microbein + 100% NPK (T12).

According to agricultural practices for growing barley in Egypt, the recommended dose of mineral fertilizers was as follows: nitrogen fertilizers were applied at a rate of 180 kg N ha^−1^, added in the form of ammonium nitrate fertilizer (33.5% N) in three equivalent parts; the first part was fertilized 21 days after planting (DAS), the second was at 35 DAS, and the third part was added at 50 DAS. As for phosphorus, it was fertilized at a rate of 62 kg P_2_O_5_ ha^−1^ in the form of calcium superphosphate fertilizer (15.5% P_2_O_5_) at a rate of 400 kg per hectare. As for potassium fertilizer, it was added at a rate of 71 kg K_2_O ha^−1^ in the form of potassium sulfate (48% K_2_O) in one dose with the first dose of nitrogen fertilizer. The seed rate used was 15.48 g per m^2^ of barley grain and sowing was carried out on 1 December for both seasons following all other cultural practices recommended for planting barley fields. Each plot had 4 ridges, where the length of the ridge was 3.50 m and its width was 70 cm. Thus, the area of the piece was 11.2 m^2^.

### 4.3. Data Recorded

#### 4.3.1. Agronomic Traits

At harvest, ten plants were randomly selected from each plot to evaluate plant height (cm), spike length (cm), spike weight (g), number of grains spike^−1^, number of spikes/m^2^, 1000 grain weight, while the biological yield, grain yield, and straw yield were evaluated per plot and then converted to grain yield (Mg ha^−1^), straw yield (Mg ha^−1^), and biological yield (Mg ha^−1^).

#### 4.3.2. Chemical Composition

According to AOAC [94], the micro-Kjeldahl method was followed and used to evaluate total N in grains and straw. The protein content of grain and straw was estimated by multiplying the total nitrogen by 4.64 [95]. The chlorostannous-reduced molybdophosphoric blue color method according to Chapman and Parker [96] was used to determine the phosphorus content (P%) colorimetrically. The flame photometric method according to Page et al. [97] was used to determine the potassium content (K%) in digested plant materials. The uptake of N, P, and K (kg ha^−1^) in grains and straw was estimated by multiplying the yield of grains or straw by their N%, P%, and K% content, respectively.

#### 4.3.3. Soil Bio-Activity

Soil samples were collected from the upper layer at a depth of 25 cm from 3 soil patches in a random manner using a drill from different replicates for each treatment. Then, the samples were collected, mixed in a homogeneous manner, and air-dried. Soil samples were drawn from this material to calculate the microbial count [98]. While the activity of dehydrogenase (DHA) enzyme in the rhizosphere soil was measured according to the method Casida et al. [99], to estimate microbial activity, the antioxidant activity in the rhizosphere soil was also evaluated after the extraction procedure carried out by Rimmer and Abbott [100].

### 4.4. Statistical Analysis

These data were statistically analyzed using analysis of variance (ANOVA) for a randomized complete block design according to Snedecor and Cochran [101] using CoStat version 6.303 [102]. Treatment means were compared by performing Duncan’s multiple range test at a 0.05 probability level [103].

## 5. Conclusions

Bio-fertilizers were applied to barley with three compounds. The first compound contains various phosphate-solubilizing bacteria, the second compound contains a variety of bacteria that fix nitrogen, and the third compound contains many species from both groups. This led to a significant improvement in all the characteristics under study, including productivity and grain yield. The results showed that the application of bio-fertilizers with the addition of 75% of the recommended dose of the mineral fertilizer was a higher production quantity than that obtained from applying 100% of the mineral fertilizer with the recommended dose of chemical fertilizers. This reduces the cost of mineral fertilizers used in barley production under saline stress and provides the nutritional requirements of the plant, which leads to an increase in the growth and productivity of the barley plant. The application of bio-fertilizers also increased the uptake and content of nutrients in straw and grains and increased the number of microorganisms in the soil, which led to an increase in enzyme dehydrogenase and antioxidants and increased availability of nutrients.

## Figures and Tables

**Figure 1 plants-13-00951-f001:**
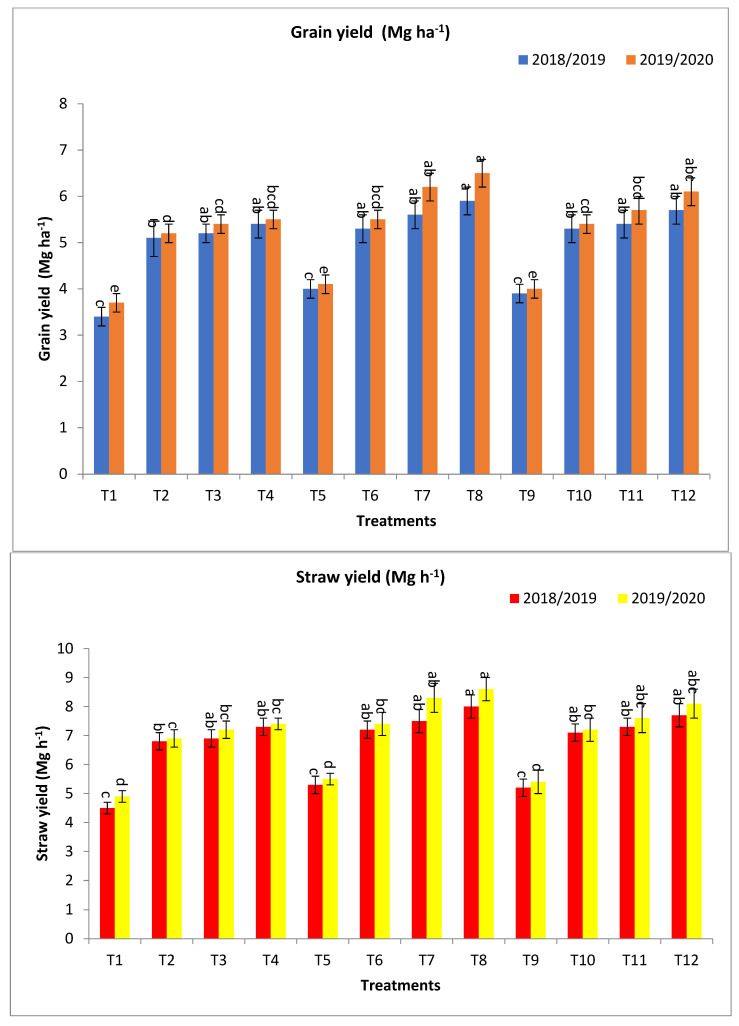
Effect of bio–fertilizers and reduction of the recommended dose of mineral fertilizers (RDP) on grain yield (Mg ha^−1^), straw yield (Mg ha^−1^), and biological yield (Mg ha^−1^) of barley under saline conditions during 2018/2019 and 2019/2020 seasons. T1: without bio-fertilizer + 50% NPK, T2: Phosphorin + 50% NPK, T3: Rhizobacterin + 50% NPK, T4: Microbein + 50% NPK, T5: without bio-fertilizer + 75% NPK, T6: Phosphorin + 75% NPK, T7: Rhizobacterin + 75% NPK, T8: Microbein + 75% NPK, T9: without bio-fertilizer + 100% NPK, T10: Phosphorin + 100% NPK, T11: Rhizobacterin + 100% NPK, and T12: Microbein + 100% NPK; bar represents the standard error, different letters next to the values point to significant differences at *p* ≤ 0.05 according to Duncan’s multiple range test.

**Figure 2 plants-13-00951-f002:**
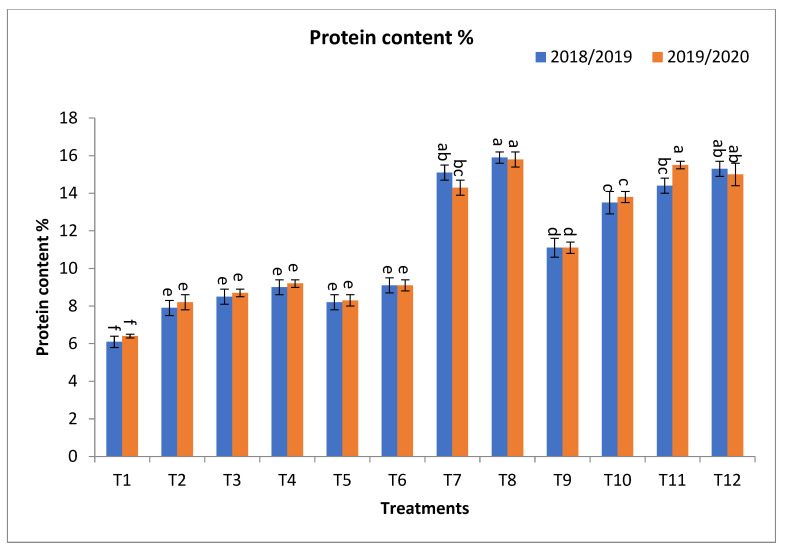
Effect of bio-fertilizers and reduction of the recommended dose of mineral fertilizers (RDP) on protein content %, Grain N, P, and K uptake (kg ha^−1^) of barley under saline conditions during 2018/2019 and 2019/2020 seasons. T1: without bio-fertilizer + 50% NPK, T2: Phosphorin + 50% NPK, T3: Rhizobacterin + 50% NPK, T4: Microbein + 50% NPK, T5: without bio-fertilizer + 75% NPK, T6: Phosphorin + 75% NPK, T7: Rhizobacterin + 75% NPK, T8: Microbein + 75% NPK, T9: without bio-fertilizer + 100% NPK, T10: Phosphorin + 100% NPK, T11: Rhizobacterin + 100% NPK, and T12: Microbein + 100% NPK; bar represents the standard error, different letters next to the values point to significant differences at *p* ≤ 0.05 according to Duncan’s multiple range test.

**Figure 3 plants-13-00951-f003:**
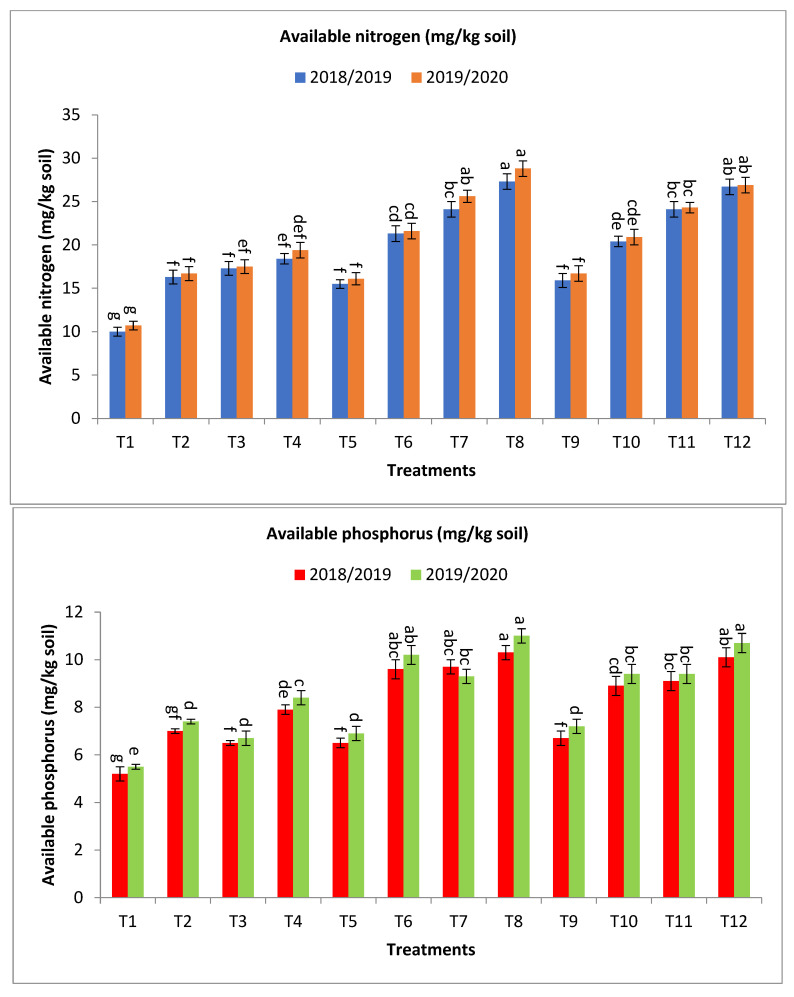
Effect of bio-fertilizers and reduction of the recommended dose of mineral fertilizers (RDP) on available nitrogen (mg/kg soil), available phosphorus (mg/kg soil), total microbial count ×10^5^ cfu/g dry soil, antioxidant activity of soil ug ascorbic acid/g dry soil (AOA), dehydrogenase (enzyme μg TPF/g dry soil/24 h), nitrogen fixers (MPN/gm dry soil) 10^3^, and phosphate-solubilizing bacteria (PDB) counts (CFU/gm dry soil) of barley under saline conditions during 2018/2019 and 2019/2020 seasons. T1: without bio-fertilizer + 50% NPK, T2: Phosphorin + 50% NPK, T3: Rhizobacterin + 50% NPK, T4: Microbein + 50% NPK, T5: without bio-fertilizer + 75% NPK, T6: Phosphorin + 75% NPK, T7: Rhizobacterin + 75% NPK, T8: Microbein + 75% NPK, T9: without bio-fertilizer + 100% NPK, T10: Phosphorin + 100% NPK, T11: Rhizobacterin + 100% NPK, and T12: Microbein + 100% NPK; bar represents the standard error, different letters next to the values point to significant differences at *p* ≤ 0.05 according to Duncan’s multiple range test.

**Table 1 plants-13-00951-t001:** The effect of treatment with bio-fertilizers and reducing mineral fertilizers by percentages of the recommended dose of mineral fertilizers (RDP) on some agronomic traits of barley variety Giza 123 in the years 2018/2019 and 2019/2020.

Treatments	Plant Height (cm)		Spike Length (cm)		Spike Weight (g)	
	2018/2019	2019/2020	2018/2019	2019/2020	2018/2019	2019/2020
T1	68.1 ± 0.4 f	72.4 ± 1.3 f	8 ± 0.4 e	8.1 ± 0.2 g	1.6 ± 0.1 d	1.6 ± 0.1 d
T2	82.1 ± 1.6 e	87.4 ± 1.6 e	9.6 ± 0.2 d	9.8 ± 0.2 f	2.2 ± 0.1 c	2.3 ± 0.1 c
T3	92.4 ± 0.5 d	98.6 ± 1.8 cd	10.8 ± 0.2 bcd	11.1 ± 0.5 def	2.4 ± 0.1 c	2.4 ± 0.1 c
T4	96.9 ± 0.4 cd	113.2 ± 1.6 ab	10.2 ± 0.3 cd	12.7 ± 0.4 abc	2.4 ± 0.1 c	2.5 ± 0.1 c
T5	83.4 ± 1.5 e	89 ± 0.8 e	9.8 ± 0.3 d	10 ± 0.5 ef	2.3 ± 0.1 c	2.3 ± 0.1 c
T6	97.5 ± 1.7 cd	105.3 ± 1.7 c	11.4 ± 0.5 bc	11.8 ± 0.4 bcd	2.4 ± 0.1 c	2.6 ± 0.1 c
T7	104 ± 0.8 b	111.8 ± 2 b	12.2 ± 0.6 ab	12.6 ± 0.6 abc	3.8 ± 0.2 a	4 ± 0.2 a
T8	112 ± 1.8 a	119.1 ± 1.6 a	13.1 ± 0.6 a	13.4 ± 0.6 a	4 ± 0.2 a	4.1 ± 0.2 a
T9	84.4 ± 2.1 e	92.5 ± 1.4 de	9.9 ± 0.5 d	10.4 ± 0.5 def	3.1 ± 0.1 b	3.1 ± 0.1 b
T10	93.3 ± 2 cd	101.3 ± 1.8 c	10.9 ± 0.5 bcd	11.4 ± 0.5 cde	3.7 ± 0.2 a	3.9 ± 0.2 a
T11	98.2 ± 1.9 c	103.1 ± 1.8 c	11.5 ± 0.5 bc	11.6 ± 0.5 ef	4.1 ± 0.2 a	4.2 ± 0.2 a
T12	110.3 ± 1.7 a	115.4 ± 1.6 ab	12.9 ± 0.6 a	13 ± 0.6 ab	4.1 ± 0.2 a	4.2 ± 0.2 a
**Treatments**	**Number of Grains Spike^−1^**		**No. of Spikes/m^2^**		**1000-Grain Weight (g)**	
	**2018/2019**	**2019/2020**	**2018/2019**	**2019/2020**	**2018/2019**	**2019/2020**
T1	25.7 ± 1.2 d	27.5 ± 1.2 d	320.7 ± 7.1 e	303.6 ± 5.4 g	25.2 ± 0.6 f	27.4 ± 0.5 e
T2	36.4 ± 1.7 c	38.8 ± 1.8 c	323.5 ± 8.7 e	317.8 ± 5.3 fg	35.8 ± 1 e	38.8 ± 0.7 d
T3	39.2 ± 1.9 c	41.2 ± 1.9 c	337.4 ± 7.2 de	349.4 ± 11.8 ef	38.6 ± 0.8 de	41.2 ± 1.4 d
T4	40.2 ± 1.9 c	42.7 ± 1.9 c	344.4 ± 10 cde	361 ± 7.3 e	39.6 ± 1.8 d	42.7 ± 0.9 d
T5	37.7 ± 1.8 c	39.1 ± 1.8 c	343.4 ± 6.7 cde	320.1 ± 9.2 fg	37.1 ± 1.8 de	39.1 ± 1.1 d
T6	40.1 ± 1.9 c	43.1 ± 2 c	376.1 ± 7.5 abc	436 ± 8.7 cd	39.5 ± 0.8 de	43 ± 0.9 d
T7	63.9 ± 1.5 a	67.3 ± 1.6 a	388.4 ± 13 ab	463.7 ± 9.6 cd	63.1 ± 1.5 b	67.4 ± 1.4 b
T8	65.7 ± 1.8 a	69.2 ± 1.8 a	410.4 ± 12.6 ab	509.1 ± 9.3 bc	64.9 ± 1.4 ab	69.3 ± 1.3 ab
T9	50.9 ± 2 b	52.6 ± 1.6 b	332.6 ± 11.5 e	339.4 ± 6.8 ef	50.3 ± 1.5 c	52.6 ± 1 c
T10	61.8 ± 1.9 a	65.3 ± 1.5 a	369.5 ± 12.4 bcd	419.4 ± 9.4 d	61 ± 1.5 ab	65.4 ± 1.6 ab
T11	68.5 ± 1.9 a	71.5 ± 1.3 a	384.5 ± 5.4 ab	444.8 ± 8.2 bcd	67.3 ± 0.9 a	71.6 ± 1.3 ab
T12	68.7 ± 2.3 a	71.6 ± 1.4 a	395.4 ± 10.3 ab	473.3 ± 11.7 b	67.8 ± 1.4 a	71.7 ± 1.4 a

T1: without bio-fertilizer + 50% NPK, T2: Phosphorin + 50% NPK, T3: Rhizobacterin + 50% NPK, T4: Microbein + 50% NPK, T5: without bio-fertilizer + 75% NPK, T6: Phosphorin + 75% NPK, T7: Rhizobacterin + 75% NPK, T8: Microbein + 75% NPK, T9: without bio-fertilizer + 100% NPK, T10: Phosphorin + 100% NPK, T11: Rhizobacterin + 100% NPK, and T12: Microbein + 100% NPK; the values are the mean values ± standard error values, different letters next to the values point to significant differences at *p* ≤ 0.05 according to Duncan’s multiple range test.

**Table 2 plants-13-00951-t002:** The effect of treatment with bio-fertilizers and reducing mineral fertilizers by percentages of the recommended dose of mineral fertilizers (RDP) on nitrogen, phosphorus, and potassium uptake of straw (kg ha^−1^) of Giza 123 barley cultivar in the 2018/2019 and 2019/2020 seasons.

Treatments	N/Straw (kg ha^−1^)		K/Straw (kg ha^−1^)		P/Straw (kg ha^−1^)	
	2018/2019	2019/2020	2018/2019	2019/2020	2018/2019	2019/2020
T1	30.9 ± 1.5 f	33.3 ± 1 f	20.1 ± 1 d	20.9 ± 1 e	8.6 ± 0.4 e	8.9 ± 0.4 f
T2	43.3 ± 2.1 e	46.6 ± 0.8 e	25.6 ± 1.2 c	26.6 ± 1.2 d	12.2 ± 0.6 cd	12.6 ± 0.6 de
T3	46.6 ± 2.2 e	49.4 ± 0.6 e	25.6 ± 1.2 c	26.8 ± 1.2 d	11.2 ± 0.5 d	12.1 ± 0.6 de
T4	47.1 ± 2.2 e	50.6 ± 0.8 e	31.9 ± 1.5 a	33.2 ± 1.5 a	13.4 ± 0.6 bc	13.7 ± 0.6 cd
T5	44.9 ± 2.1 e	47 ± 1.2 e	24.9 ± 1.2 c	26 ± 1.2 d	10.5 ± 0.5 de	11 ± 0.5 e
T6	47.7 ± 2.3 e	51.6 ± 1.6 e	25.8 ± 1.2 c	26.9 ± 1.3 d	16.7 ± 0.8 a	16.8 ± 0.8 ab
T7	75.3 ± 3.5 bc	80 ± 2.9 c	26.8 ± 1.3 c	27.9 ± 1.3 cd	15.6 ± 0.7 a	15.8 ± 0.7 ab
T8	85.7 ± 2.9 a	86.9 ± 2.8 a	32.1 ± 1.6 a	32.8 ± 1.5 ab	17.1 ± 0.8 a	17.4 ± 0.8 a
T9	60.3 ± 2.9 d	62.8 ± 2.9 d	26.6 ± 1.3 c	27.7 ± 1.3 cd	11.2 ± 0.5 d	11.6 ± 0.5 e
T10	72.9 ± 3.5 c	77.7 ± 3.6 c	28.3 ± 1.4 abc	29.5 ± 1.4 bcd	16 ± 0.8 a	15.8 ± 0.7 a
T11	82.5 ± 3.9 ab	85.5 ± 3.2 ab	27.5 ± 1.3 bc	28.6 ± 1.3 cd	15.3 ± 0.7 ab	15.1 ± 0.7 bc
T12	83.9 ± 4 a	85.6 ± 2.9 ab	31.3 ± 1.5 ab	31.3 ± 0.9 abc	16.8 ± 0.8 a	16.8 ± 0.8 ab

T1: without bio-fertilizer + 50% NPK, T2: Phosphorin + 50% NPK, T3: Rhizobacterin + 50% NPK, T4: Microbein + 50% NPK, T5: without bio-fertilizer + 75% NPK, T6: Phosphorin + 75% NPK, T7: Rhizobacterin + 75% NPK, T8: Microbein + 75% NPK, T9: without bio-fertilizer + 100% NPK, T10: Phosphorin + 100% NPK, T11: Rhizobacterin + 100% NPK, and T12: Microbein + 100% NPK; the values are the mean values ± standard error values, different letters next to the values point to significant differences at *p* ≤ 0.05 according to Duncan’s multiple range test.

**Table 3 plants-13-00951-t003:** Physical and chemical properties of pre-cultivated soil and irrigated water for the experiment site.

Sample Depth (cm)				Physical Properties %						Soil Texture		
0–30				**Sand**	**Silt**	**Clay**				**Sandy Silt Loam**		
				51.95	25.73	22.32						
**Chemical analysis**												
**Cation meq/L**									**Anion meq/L**			
pH	E.C ds.m^−1^	CaCo_3_%	O.M %	T.N %	Ca^+2^	Mg^+2^	Na^+^	K^+^	Co3^−2^	HCo_3_	Cl^−^	So_4_^−2^
8.67	8.24	4.29	0.56	0.09	25.2	7.83	44.35	5.34	0.92	15.67	48.9	5.92
**Chemical analysis of irrigation water**												
**Cation meq./L**							**Anion meq./L**					
pH	E.C ds.m^−1^	Ca^+2^	Mg^+2^	Na^+^	K^+^		CO_3_^−2^	HCO_3_^−^		Cl^−^	SO_4_^−2^	
7.62	2.34	5.37	3.12	5.43	0.25		1.63	12.37		4.23	6.18	

## Data Availability

Data are contained within the article.

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
