# Peer review of "Effect of Bio-Fertilizer Application on Agronomic Traits, Yield, and Nutrient Uptake of Barley (Hordeum vulgare) in Saline Soil"

_plants, 2024, doi:10.3390/plants13070951_

Round 1
Reviewer 1 Report
Comments and Suggestions for Authors
Overall, the manuscript is well written. The introduction and methodology are clearly stated.
However, the statistical analysis is required to be reprocessed as in the tables and figures there are no significance letters involved. Please provide clear statistical analysis for all data to demonstrate differences between treatments. Very difficult to follow which treatments are better (only values, no letters). Too many numbers (e.g., percentage increase rate or highest value, etc.) in the result section, so please present the results in a concise and organized manner.
In the discussion section, it will be good to discuss the potential mechanisms by which N-fixing and phosphate-solubilizing bacteria may alleviate salinity stress and enhance nutrient uptake in barley.
In the conclusion section, the authors state that the application of bio-fertilizers with 75% NPK was a higher production than that of 100% NPK; this reduced the cost of chemical fertilizer application under saline stress. But, has the cost calculation been carried out in the supplementary as the cost for bio-fertilizer is not cheap either. It will be good to have this supporting information based on current market price.
Author Response
The responses to Review Report (Reviewer 1)
Overall, the manuscript is well written. The introduction and methodology are clearly stated.
However, the statistical analysis is required to be reprocessed as in the tables and figures there are no significance letters involved. Please provide clear statistical analysis for all data to demonstrate differences between treatments. Very difficult to follow which treatments are better (only values, no letters).
Done
Too many numbers (e.g., percentage increase rate or highest value, etc.) in the result section, so please present the results in a concise and organized manner.
Done
In the discussion section, it will be good to discuss the potential mechanisms by which N-fixing and phosphate-solubilizing bacteria may alleviate salinity stress and enhance nutrient uptake in barley.
Response: Thank you very much, in the discussion section, we tried to discuss the potential mechanisms by which nitrogen-fixing and phosphate-solubilizing bacteria may alleviate salinity stress and enhance nutrient uptake in barley through the following paragraphs:
Line 577-line 579 : Treatment and inoculation with PGP bacteria led to enhancing and increasing plant growth under salinity conditions through the production of growth-promoting nutrients and growth regulators [82].
592-599 : Under saline conditions, bio-fertilizers promoted plant growth by facilitating enhanced nutrients and produce growth-promoting substances and growth regulators [37,83,84]. The reason for the increase in yield and its components in treatments received with bio-fertilizers is mainly due to the beneficial effect of applying bio-fertilizers to the soil, as this improved the biological, and chemical and physical properties of the soil. All of these effects resulted in an increase in the availability and release of more nutrients available to the roots plants [85-87].
Line 629 – line 637: The results showed that bio-fertilizers have the ability to enhance plant growth, due to their ability to solubilize phosphate and produce organic acids, exopolysaccharides, and IAA. Therefore, microorganisms with PGP activities can lessen biotic and abiotic stresses, thus being useful in agricultural fields [103-105]. PGP microbes improve stress tolerance in plants through various mechanisms; such as the production of antioxidants that can detoxify reactive oxygen species, which leads to improved growth standards, the production of growth regulators, and most importantly, increased nutrient content through nitrogen fixation and dissolution of elements. All of these mechanisms lead to improving plant health under salt stress [82,106,107].
Line 684-690: Bio-fertilizers containing PGPR bacteria play an necessary role in providing and cycling nutrients in the soil through several processes, including that they participate in processes such as nitrogen fixation, nitrification, oxidation, ammonification, and other processes that lead to the decomposition of organic matter in the soil and the release of vital inorganic plant nutrients in the soil [111-113]. Kurokura et al. [114], shown that the secretion of organic acids by PGPR contributes significantly to increasing the solubility of phosphorus and converting it from insoluble to soluble forms [115-117].
Line 697 – line 705: On the other side, over time, bio-fertilizer has proven to be an effective means of providing the necessary nutrients to plants and maintaining their the sustainability of the agricultural system and the ecosystem in general. Bio-fertilizers consist of a variety of beneficial microorganisms that are able to decompose essential nutrients from insoluble compounds, making them available to plants. As a result of successive studies on this, there are a large number of microorganisms, including bacteria, fungi, and actinomycetes, that have properties that enhance the dissolution of metal ions through various mechanisms, such as changing the pH of the soil or direct heavy metal removal from metal cations [118-120].
Line 740 – line 754: The antioxidant system is a very useful indicator of soil quality and the biological condition of the soil [134-136]. The dehydrogenase enzyme increases with the addition of bio-fertilizers with the increase in the addition of mineral fertilizers, but it increases to a certain extent and not a continuous increase. In the same direction and with regard to the effect of bio-fertilizers, the results showed an increase in the dehydrogenase enzyme when adding bio-fertilizers in the presence of the same rate of mineral fertilizers, with a relative decrease in the rate greater than 100%. The results indicated that the dehydrogenase enzyme was higher when using 75% NPK/ha compared to that obtained at 50% NPK/ha or 100% NPK/ha, (Figure 3). Increased antioxidant activity in the soil protects the plant from oxidative stress resulting from high salinity [137,138]. Enzymatic and non-enzymatic antioxidants play an essential function in plant development and growth, especially in protection mechanisms [139-141]. Growth of plants under environmental stresses leads to oxidative stress, causing a pathological condition. Numerous studies have confirmed the existence of a strong relationship between the number of microbes and the total antioxidant capacity in soil and in different soil types [142,143].
- In the conclusion section, the authors state that the application of bio-fertilizers with 75% NPK was a higher production than that of 100% NPK; this reduced the cost of chemical fertilizer application under saline stress. But, has the cost calculation been carried out in the supplementary as the cost for bio-fertilizer is not cheap either. It will be good to have this supporting information based on current market price.
Response:
N : 7.75 × 775 = 6000 LE
P : 8 × 175 = 1400 LE
K : 3 × 2100 = 6300 LE
Sum. = 13700 × 0.25 = 3425 LE
Saving on the cost of chemical fertilizers = 13700 × 0.25 = 3425
Phosphorein : 1.4 kg x 35 = 122.5 LE
Rhizobacterin : 2.4 kg x 35 = 210 LE
Microbein : 1 kg x 35 = 87.5 LE
Sum. = 122.5 + 210 + 87.5 = 420
Net savings in the cost of chemical fertilizers = 3425 – 420 = 3005 LE

Reviewer 2 Report
Comments and Suggestions for Authors
The manuscript plants-2898754 entitled "Nitrogen fixing bacteria, phosphate solubilizing bacteria, and their mixtures to reduce mineral fertilizers and improve yield, agronomic traits and nutrient uptake in barley under salinity stres" describe an interesting experimental activity regarding the use of PGP to support barley plant under saline stress conditions.
Title: It is too long. I sugest to the authors to reduce its lengh.
Abstract: Needs some revisions
Keyowords: are fine
Introduction: It is too long and dispersive. It should describe the state of the art within 1.5 pages. Revise according my specific comments.
M&M: Should be revised and implemented
Results: The results of statistical analysis are missing. The paragraph should be implemented and semplified.
Discussion: The first part is pretty speculative and could be deleted. The second part is fine. In the discussion try to explaing the observed results
Conclusions: Clear and based on the observed results
My specific comments are enclosed in the attached pdf file.

The manuscript should be proof readed
Author Response
The responses to Review Report (Reviewer 2)
Comments and Suggestions for Authors
The manuscript plants-2898754 entitled "Nitrogen fixing bacteria, phosphate solubilizing bacteria, and their mixtures to reduce mineral fertilizers and improve yield, agronomic traits and nutrient uptake in barley under salinity stres" describe an interesting experimental activity regarding the use of PGP to support barley plant under saline stress conditions.
Title: It is too long. I sugest to the authors to reduce its lengh.
It has been changed as suggested:
Effect of Biofertilizer Application on Agronomic Traits, Yield and Nutrient Uptake of Barley (Hordeum vulgare) in Saline Soil
Abstract: Needs some revisions
Done
Keyowords: are fine
Thank you
Introduction: It is too long and dispersive. It should describe the state of the art within 1.5 pages. Revise according my specific comments.
Done
M&M: Should be revised and implemented
Results: The results of statistical analysis are missing. The paragraph should be implemented and semplified.
Done
Discussion: The first part is pretty speculative and could be deleted. The second part is fine. In the discussion try to explaing the observed results
Done
Conclusions: Clear and based on the observed results
Thank you
Respond to specific comments attached in the attached pdf file.
Line 163 – line 166 : made appropriate citation of trademark
Done and changed to : Phosphorein, Rhizobacterin and Microbein
Line 164: for the commercial products please indicate the manufacturer.
Done and added :
The biofertilizers were obtained from General Organization for Agricultural Equalization Fund (GOAEF), Egypt’s Ministry of Agriculture and Land Reclamation’s.
Line 167: was: It has been changed to: were
Line 171: Done, changed to
The main properties regarding soil and water used in the experiment are detailed in the Table 1.
Line 180 : 180 kg N ha−1
this is a huge amount for the barley. Please, justify.
Response:
The doses of NPK fertilization mentioned in the manuscript are according to the recommended doses of NPK fertilization for barley in sandy soil and new reclaimed lands according to the Egyptian Ministry of Agriculture.
Line 192: 154.76 kg ha-1
it would be better to specify the investment per m2
Line 199: ten plants : its a very small sample
Response:
A number of ten plants were used to estimate the traits : plant height (cm), spike length (cm), spikes weight (g), number of grains spike−1, number of spikes/m2, 1000-grain weight, while the biological yield, grain yield, and straw yield were evaluated per plot and then converted to grain yield (ton ha-1), straw yield (ton ha-1), and biological yield (ton ha-1).
Corrected to :
"At harvest, ten plants were randomly selected from each plot to evaluate plant height (cm), spike Length (cm), spike weight (g), number of grains spike−1, number of spikes/m2, 1000-grain weight, while the biological yield, grain yield, and straw yield were evaluated per plot and then converted to grain yield (ton ha-1), straw yield (ton ha-1), and biological yield (ton ha-1)".
line 211: was determined
Done: it deleted
Line 215: Please add a description of sampling.
Done
Soil samples were collected from the upper layer at a depth of 25 cm from 3 soil patches in a random manner using a drill from different replicates for each treatment. Then the samples were collected, mixed in a homogeneous manner, and air-dried. Soil samples were drawn from this material to calculate the microbial count (Allen, 1959).
Line 222: Why did you use Monte Carlo analysis?
Response
I'm sorry for the error, Casella's method is mentioned here as a statistical basis for Costa version 6.303.
As an error and has been corrected to:
Snedecor, G.W.; Cochran, W.G. Statistical Methods, 5th ed.; Iowa State University: Press, IA, USA, 1967.
Line 225: Results of statistical analysis are not present in the MS.
Done
Line 239 : Here and in whole manuscript. One decimal is enough for percentage values
Done
Line 253: Table 1.
I suggest providing acronyms that are easier to understand and include in the tables and graphs
Response: Some of these acronyms have been changed:
Rhizobacterin: Rhizo.;
Phosphorein: Phospho.;
Microbein: Micro.;
There are also some other possible acronyms. What's the easiest one?
Rhizobacterin: Azoto. NF: nitrogen fixation
Phosphorein: PDB. PS: phosphate-solubilizing
Microbein: Mixt. NF and PS : mixture of nitrogen fixation and
phosphate-solubilizing bacteria
Line 254: where are the letters?
Letters have been added
Line 265: here and in the whole manuscript. Spikes
Done
Line 294: The data presented in figures A, B, C should be included in the previous tables.
Done
The data presented in Figure 1 Figures A, B, and C are included in Table 2.
Line 310: units should be Mg ha-1
Done
Line 312: do not repeat data already present in the figure or table
Done
Line 449: units?
Done
Line 649: avoid to repeat the results in this section.
Done
Comments on the Quality of English Language
The manuscript should be proof readed
Done

Reviewer 3 Report
Comments and Suggestions for Authors
It is interesting and useful that the authors have investigated the effect of bio-fertilizer application on agronomic traits, yield and nutrient uptake of barley (Hordeum vulgare) in saline soil condition. In total, the MS was written sound. Hence, it is recommended to be accepted after some revisions.
1. Shortened the title, such as “effect of biofertilizer application on agronomic traits, yield and nutrient uptake of barley (Hordeum vulgare) in saline soil”;
2. Give some reasons why the three biofertilizers were used at the different rates (Rhizobacterien 2.4 kg, Phosphorien 1.4 kg, Mycrobein 1 kg per 143 kg barley grains ha−1);
3. “two field experiments were conducted to study…..” was not clearly presented. Not sure if experiments were conducted in the same field or not?
4. It is better to use sub-sections for different outcomes in the section of Results.
5. It was not presented any “Different letters next to the values”, which pointed to significant differences at p ≤ 0.05 according to Duncan’s multiple range test in Table 1 & 2.
6. Changed “the values” into “different letters” in all figures.
7. It is better to use sub-sections in the section of Disccusions.
Comments on the Quality of English LanguageMinor editing of English language is required.
Author Response
The responses to Review Report (Reviewer 3)
Comments and Suggestions for Authors
It is interesting and useful that the authors have investigated the effect of bio-fertilizer application on agronomic traits, yield and nutrient uptake of barley (Hordeum vulgare) in saline soil condition. In total, the MS was written sound. Hence, it is recommended to be accepted after some revisions.
- Shortened the title, such as “effect of biofertilizer application on agronomic traits, yield and nutrient uptake of barley (Hordeum vulgare) in saline soil”;
It has been changed as suggested:
Effect of Biofertilizer Application on Agronomic Traits, Yield and Nutrient Uptake of Barley (Hordeum vulgare) in Saline Soil
- Give some reasons why the three biofertilizers were used at the different rates (Rhizobacterien 2.4 kg, Phosphorien 1.4 kg, Mycrobein 1 kg per 143 kg barley grains ha−1);
Response:
- The three biofertilizers were applied at different rates (Rhizobacterien 2.4 kg, Phosphorien 1.4 kg, Mycrobein 1 kg per 143 kg barley grain ha−1); According to the producer of these fertilizers, which is the Egyptian Ministry of Agriculture.
- Also, due to the difference in the amount of carrier material in each of the three biofertilizers
- “two field experiments were conducted to study…..” was not clearly presented. Not sure if experiments were conducted in the same field or not?
Response:
- The experiments were conducted at the same site
- It is better to use sub-sections for different outcomes in the section of Results.
Done
- It was not presented any “Different letters next to the values”, which pointed to significant differences at p ≤ 0.05 according to Duncan’s multiple range test in Table 1 & 2.
Letters have been added
- Changed “the values” into “different letters” in all figures.
Done, letters have been added
- It is better to use sub-sections in the section of Disccusions.
Done

Round 2
Reviewer 1 Report
Comments and Suggestions for Authors
Authors relatively addressed the listed issues.
Authors can use T1, T2, T3, ....T12 to replace the treatments across the entire manuscript including tables, figures, etc. which will make the paper nice and neat.
Comments on the Quality of English Languagegood
Reviewer 2 Report
Comments and Suggestions for Authors
The manuscript was improved according with my suggestion in all its parts.
Reviewer 3 Report
Comments and Suggestions for Authors
This is a recised MS. The authors have corrected reviewers' suggestions. Hence, it is recommended to be accepted in the present form.
Comments on the Quality of English LanguageMinor editing of English language is required.